# Gallbladder Cancer: Current Multimodality Treatment Concepts and Future Directions

**DOI:** 10.3390/cancers14225580

**Published:** 2022-11-14

**Authors:** Niklas Sturm, Jasmin Selina Schuhbaur, Felix Hüttner, Lukas Perkhofer, Thomas Jens Ettrich

**Affiliations:** 1Department of Internal Medicine I, Ulm University Hospital, 89081 Ulm, Germany; 2Department of General and Visceral Surgery, Ulm University Hospital, 89081 Ulm, Germany

**Keywords:** gallbladder cancer, targeted therapy, multidisciplinary cancer therapy

## Abstract

**Simple Summary:**

Gallbladder cancer is the most common malignancy of the biliary tract and is associated with several risk factors such as female sex, ethnic background, and chronic biliary inflammation. If gallbladder cancer is detected in a localized stage, it can be cured by surgical resection. In advanced-stage disease, chemotherapy can render up to one-third of patients eligible for surgery with better prognosis. For high-risk patients, chemotherapy after surgery can also help to prevent disease recurrence. In non-resectable patients, a chemotherapy regimen of gemcitabine and cisplatin is the current standard of care but might be soon extended by immunotherapy with durvalumab. Since the unique tumor biology of gallbladder cancer harbors the opportunity for molecular targeted therapy approaches, current research has focused on new therapeutic agents that might improve the poor prognosis of advanced disease.

**Abstract:**

Gallbladder cancer (GBC) is the most common primary tumor site of biliary tract cancer (BTC), accounting for 0.6% of newly diagnosed cancers and 0.9% of cancer-related deaths. Risk factors, including female sex, age, ethnic background, and chronic inflammation of the gallbladder, have been identified. Surgery is the only curative option for early-stage GBC, but only 10% of patients are primary eligible for curative treatment. After neoadjuvant treatment, up to one-third of locally advanced GBC patients could benefit from secondary surgical treatment. After surgery, only a high-risk subset of patients benefits from adjuvant treatment. For advanced-stage GBC, palliative chemotherapy with gemcitabine and cisplatin is the current standard of care in line with other BTCs. After the failure of gemcitabine and cisplatin, data for second-line treatment in non-resectable GBC is poor, and the only recommended chemotherapy regimen is FOLFOX (5-FU/folinic acid and oxaliplatin). Recent advances with the PD-L1 inhibitor durvalumab open the therapy landscape for immune checkpoint inhibition in GBC. Meanwhile, targeted therapy approaches are a cornerstone of GBC therapy based on molecular profiling and new evidence of molecular differences between different BTC forms and might further improve the prognosis of GBC patients.

## 1. Introduction

Gallbladder cancer (GBC) is taxonomically described as a subgroup of biliary tract cancer (BTC), along with intrahepatic cholangiocarcinoma (iCCA), extrahepatic cholangiocarcinoma (eCCA), perihilar cholangiocarcinoma (pCCA or Klatskin tumor), and distal cholangiocarcinoma (dCCA) [1,2].

Gallbladder cancer (GBC) is the most common entity of malignant biliary tract cancers (BTC) [1]. As one of the most aggressive malignant tumors, GBC is accompanied by high lethality (median 5-year overall survival 18%) [2]. In 2020, 19.3 million cancer diagnoses and almost 10 million cancer-related deaths occurred worldwide, GBC accounting for 0.6% of newly diagnosed cancers and 0.9% of cancer related deaths [3]. The decreasing incidence of GBC in the Western world is believed to be connected to the rising number of routine cholecystectomies [4]. Early stages of GBC usually remain asymptomatic, are potentially curable, and often manifest incidentally during or after surgery [5,6]. Unfortunately, only 10% of GBC cases are considered to be resectable at initial diagnosis [7]. Advanced stages of GBC present with jaundice, weight loss, or abdominal pain and exhibit an early metastasis pattern, resulting in a dismal prognosis [6,8]. Advanced stages of GBC are treated with systemic chemotherapy, in neoadjuvant, perioperative, adjuvant, or palliative intention. As the understanding of GBC molecular pathogenesis progresses, targeted treatment options are becoming increasingly promising.

Up till now, the understanding of GBC pathophysiology is limited, thereby impeding preventive and therapeutic measures [6]. As a complex malignancy, GBC comprises environmental, lifestyle, and genetic risk factors [6].

### 1.1. Lifestyle Risk Factors

The female gender is known as an independent risk factor of GBC with a female-to-male ratio of 3:1 [9,10]. Age > 65 years further increases the risk of falling ill with GBC [6]. Common diseases like asymptomatic cholelithiasis and obesity exponentiate the risk of developing GBC [11,12]. While 70–90% of GBC patients present with gallstones, only approximately 1% of patients presenting with gallstones will develop GBC [13]. Hypothetically, tumorigenesis is due to a chronic irritation of the gallbladder’s mucosa [14].

### 1.2. Anatomical Risk Factors

Moreover, diseases of the biliary tract, such as primary sclerosing cholangitis (PSC), chronic biliary tract infection (viral, such as hepatitis B or C, or bacterial, such as Salmonella typhi) increase the risk of all entities of BTC [12,15]. In terms of PSC, Said et al. found that 3.5% of PSC patients developed GBC [16]. Anatomic variants such as pancreaticobiliary maljunction (PBM), which is characterized by the localization of the pancreaticobiliary junction outside of the duodenal wall, can further promote the carcinogenesis of GBC by mucosal injury through the reflux of pancreatic fluid stemming from its proteolytic enzymes [17]. When PBM is accompanied by biliary dilatation, 6.8% of patients may develop GBC. Alarmingly, in cases of PBM without biliary dilatation, the risk is even more increased (35.3%) [18].

### 1.3. Geographical Risk Factors

Globally, the incidence of GBC does not pass 10/100,000. However, it is highly increased for inhabitants of the Andean region in South America, as well as North American Indians and Mexican Americans [19]. Chile represents the highest-risk area of the world for GBC with an incidence rate of ∼25/100,000 in women and ∼9/100,000 in men [9]. In Asia, northern India encounters the highest incidence of GBC (21.5/100,000), followed by Pakistan and Japan [6]. In Europe, the highest incidences of GBC occur in the East, such as Poland (14/100,000), Slovakia, and the Czech Republic [19].

### 1.4. Genetic Alterations

An abundance of genetic alterations is currently being investigated for their role in carcinogenesis or as potential treatment targets of GBC (see below). Potentially interesting genes and gene products are summarized in Figure 1. Since major geographical and etiological differences exist, the frequency of specific genetic alterations differs between GBC patients regarding particular patient subgroups and ethnical background. Furthermore, molecular alterations that already have an impact on current targeted therapy approaches are further described in the therapy section.

In a systematic review, Kuipers et al. investigated the frequency of different molecular alterations in gallbladder cancer based on 62 articles containing 3893 GBC samples that have been analyzed [20].

The authors found TP53 (tumor protein 53) to be the most frequently mutated gene in approximately 57% (range 4–71%) of all GBC patients. TP53 is a tumor suppressor gene, which is capable of inhibiting the cell-cycle progression of damaged cells and may even trigger their apoptosis. Mutations in TP53 cause the inactivation of TP53 thereby disabling its antiproliferative function [25]. Other common genetic alterations in GBC patients were mutations of SHH (sonic hedgehog gene; 20%, based on one report by Dixit et al. [21]), ELF3 (E74 Like ETS transcription factor 3; 18.6%, range 13–20%), ARID1A (AT-rich interactive domain-containing protein 1A; 14%, range 4–25%), and SMAD4 (SMAD family member 4; 13.1%, range 0–33%).

Additional common mutations that have been described in GBC patients are mutations of EGFR (epidermal growth factor receptor, synonym: Her1, ERBB1) in approximately 12% (range 4–20.6%) [22,23], ERBB2 (human epidermal growth factor receptor 2; synonym: her2/neu) in approximately 10% (range 4–16%) [22,23], and KRAS (kirsten rat sarcoma virus) in 10.3% (range 0–59%) [20].

Other mutations that are especially interesting in terms of therapeutic targets are unfortunately rare in GBC. For example, IDH1/2 (isocitrate dehydrogenase 1 and 2), which is a promising target in patients with iCCA, is only altered in 5.9% of all GBC (range 0–11.8%) [22,23]. Other more rare mutations that occur in GBC are mutations of FGFR 1–3 (fibroblast growth factor receptor; approximately 3%) [24], NTRK1–3 (neurotrophic receptor tyrosine kinase; approximately 4%) [24], and HER3 (human epidermal growth factor receptor 3; synonym: ERBB3; approximately 9.9%, range 8–11.8%).

Microsatellite-instability (MSI) is described in approximately 8–12% of GBC, leading to a dysfunctional DNA mismatch repair [23].

This review aims to give an overview of current multimodal treatment options for gallbladder cancer patients in the current state of a transforming therapy landscape. Furthermore, future directions especially regarding tailored therapy approaches that will gain importance in the near future will be discussed.

## 2. Therapy

GBC is the most common form of biliary tract cancer [26,27]. It is characterized by unique pathogenetic and molecular features that differ from other biliary tract cancer forms (e.g., inflammation-based carcinogenesis and distinctive molecular alterations, strong association with female sex, and geographical clustering) [27,28,29]. Therefore, differentiated therapy is mandatory to improve patients’ outcome and survival, especially regarding characteristic molecular alterations that bare the opportunity for the use of new targeted therapeutics.

### 2.1. Neoadjuvant Therapy

In accordance with other forms of BTC, surgical resection is the only potentially curative therapy. Since GBC is often characterized by an aggressive tumor biology [27], neoadjuvant therapy is not recommended for patients with the possibility of primary R0-resction [30]. Unfortunately, literature estimated that only 10% of all patients diagnosed with GBC present at an early stage with the opportunity for primary curative resection [25,31] with a higher rate for incidental GBC after cholecystectomy [32].

In contrast with these findings, a large retrospective analysis of 6437 patients with GBC from the SEER (Surveillance, Epidemiology, and End Results) registry between 1988 to 2003 identified 2835 patients (44%) that underwent resection in a curative intention [33]. However, it must be admitted that most of these patients had a locally advanced tumor stage with nodal invasion or unknown nodal status, leading to a bad prognosis.

According to GLOBOCAN 2018 data from the US, 43% of GBC are found after cancer had spread to regional organs or lymph nodes, while 42% were found after spreading to distant organs or lymph nodes [34].

Therefore, neoadjuvant concepts to archive secondary resectability after tumor downstaging are a focus of current research.

At present, a chemotherapy regimen based on gemcitabine and cisplatin in accordance with the ABC-02 phase III trial (*n* = 410) is recommended for the treatment of primary unresectable GBC [25]. This recommendation is based on an objective response rate (ORR) of 37.7% and a disease control rate (DCR) of 85.2% in the subgroup of GBC patients with doublet therapy (*n* = 61) compared to a ORR of 21.4% and a DCR of 76.8% with gemcitabine alone (*n* = 56).

In a systematic review of neoadjuvant chemotherapy or radio chemotherapy for advanced-stage gallbladder cancer, Hakeem and colleagues could identify eight studies that have investigated this issue with different therapeutic concepts (*n* = 474) [35]. Despite the poor quality of the included studies, their analysis showed that 50.4% of all pretreated GBC patients (*n* = 239) were eligible for curative-intended surgery after neoadjuvant therapy. In total, 191 patients (40.3%) received a curative resection after neoadjuvant therapy with a R0-rate of 92.5% (*n* = 160). These results demonstrate that up to one-third of all advanced-stage GBC patients could benefit from a neoadjuvant treatment in case of overall survival (OS) and disease-free survival (DFS). The OS significantly differed between patients who received surgery after neoadjuvant treatment (18.5–50.1 months) compared to those who were not eligible for surgery after neoadjuvant treatment (5.0–10.5 months). At the moment, the phase III GAIN-trial aims to prospectively recruit 333 patients with locally advanced BTC or incidental GBC, which will be randomized in a study arm with perioperative chemotherapy with gemcitabine and cisplatin and another study arm with immediately surgery and the option for adjuvant chemotherapy (NCT03673072) [36]. Other trials that are currently recruiting participants for the evaluation of neoadjuvant treatment in GBC or BTC including GBC are summarized in Table 1.

A potential regimen for neoadjuvant radiochemotherapy was investigated by Engineer et al. in 28 patients with locally advanced BTC [37]. Patients received 57 Gy over twenty-five fractions in the gross tumor and 45 Gy over twenty-five fractions in the surrounding nodes in combination with gemcitabine (300 mg/m^2^/week for 5 weeks). Twenty patients (71%) achieved partial or complete radiological response, and fourteen patients (56%) achieved R0 resection after pretreatment. In accordance with previous studies, R0-resected patients had a 5-year OS of 47% compared to a 5-year OS of 24% in the whole group.

Although there are different approaches for neoadjuvant treatment, the more favorable regimen (chemotherapy alone vs. radiochemotherapy) remains unclear.

In a retrospective analysis, Verma et al. investigated the treatment and outcome of 1199 patients with locally advanced, non-metastatic gallbladder cancer from the National Cancer Database [38]. Comparing patients who received chemotherapy alone (*n* = 872) with patients who received chemoradiation (*n* = 327), the latter were found to have an improved OS. Contrary to this analysis, the prospective FFCD 9902 phase III trial, which investigated the effect of radio chemotherapy (50 Gy in 25 fractions, 5 FU 300 mg/m^2^ per day 5×/week, for 5 weeks and cisplatin 20 mg/m^2^ per day, day 1 to 4 and from day 29 to 32) compared to a combined chemotherapy regimen with gemcitabine (1000 mg/m^2^, day 1 and 4, repetition every 14 day) and oxaliplatin (100 mg/m^2^, day 1 and 4, repetition every 14 days) in 34 patients with locally advanced biliary tract cancer (4 GBC) showed different results [39]. The authors found a better OS and PFS in the chemotherapy group (PFS 5.8 vs. 11.0 months, OS 13.5 vs. 19.9 months) and suggested a preference for a combined chemotherapy regimen, although limitations considering the study population size must be admitted. In conclusion, data on the efficacy of neoadjuvant chemoradiation compared to a combined chemotherapy regimen are still mandatory, especially respecting the differences between anatomic and molecular BTC subgroups.

Considering the current research in targeted therapy options and new cytostatic chemotherapy regimens (see below), promising approaches in a palliative setting might improve disease control and secondary resectability in patients with primary non-resectable GBC.

### 2.2. Surgery

GBC is most discovered incidentally during elective or emergency cholecystectomy. In contrast to non-incidental GBC, these cases usually present in an earlier stage and are thus amenable to secondary resection with curative intent, if metastatic disease is ruled out. In general, simple cholecystectomy is adequate for T1a GBC and results in excellent 5-year survival rates [40], but extended oncologic resection is recommended for tumors of stage T1b or higher [25]. While the index cholecystectomy with incidental finding of GBC is often performed at smaller hospitals, patients should be referred to a center with sufficient expertise in hepatobiliary surgery for the secondary oncologic resection [41]. The optimal extent of resection is still debated, but in general, three aspects should always be considered (1) the resection of the adjacent liver parenchyma (i.e., the gallbladder fossa/segments IVb and V), (2) locoregional lymphadenectomy, and (3) the re-resection of the cystic duct, conceivably with the resection of the common bile duct.

Regarding the extent of liver resection, there is no consensus yet, in particular for T2 tumors, whether an atypical resection of a 2–3 cm wedge of segments IVb/V is sufficient or whether an anatomical resection of these segments should be performed. The location of the tumor on the peritoneal side (T2a) or hepatic side (T2b) may influence the decision to perform an atypical or an anatomical hepatectomy of segments IVb/V, but most studies have failed to demonstrate a survival difference between both strategies, and high-quality evidence in support of either approach is lacking [42]. On the other hand, patients with non-incidental GBC usually present with more advanced tumors (i.e., T3 or T4) and thus require more extended resections. While for small T3 tumors located in the corpus or fundus of the gallbladder, an anatomical resection of segments IVb/V en bloc with cholecystectomy might be sufficient, most T3/T4 tumors can only be resected curatively by more radical procedures. Advanced tumors located in the infundibulum of the gallbladder are particularly difficult to treat surgically, and an (extended) right hepatectomy, often including vascular resection, bile duct resection, or the resection of adjacent organs, such as the duodenum, pancreatic head, or the right transverse colon, is usually necessary to achieve clear margins. While some studies, especially from Asia, have reported favorable results for such extended resections, e.g., including right hepatectomy with simultaneous partial pancreatoduodenectomy [43], most centers are more reluctant regarding such radical approaches due to the high risk of morbidity and mortality and the questionable oncologic benefit [44,45]. Consequently, such extended resections should only be performed in selected patients, e.g., young patients with good performance status and a low perioperative risk profile. Furthermore, these patients are candidates for multimodal treatment strategies, preferably within clinical trials (see above).

While the risk for lymph node metastases in T1a GBC is well below 5%, it is approximately 10–15% in T1b GBC and further increases with advanced stages [40]. Since, lymph node spread is one of the most crucial prognostic factors [46], a locoregional lymphadenectomy is mandatory in all resectable GBC of stages T1b or higher. While nearly all authors and guidelines agree that an adequate lymphadenectomy should include the lymph nodes at the cystic duct and in the hepatoduodenal ligament, there is more controversy whether it should extend to the common hepatic artery and celiac trunk and to the superior posterior pancreaticoduodenal lymph nodes [25,47]. The aortocaval lymph nodes may be sampled in cases with suspected lymph node involvement in this location, but if positive, the prognosis is poor, similar to distant metastatic disease, and thus extended resection should be critically reconsidered [48]. Whereas the anatomical extent of lymphadenectomy is still debated, it is nowadays widely accepted that optimally at least six lymph nodes should be resected, on one hand for adequate staging and on the other hand due to a potential survival benefit for patients with N0 disease based on the assessment of six nodules compared to N0 based on fewer lymph nodes [49].

Regarding bile duct resection, especially in cases with unclear resection status during the index cholecystectomy, the cystic duct stump should always be re-resected during secondary oncologic resection and should be sent for frozen section. In case of margin positivity, a resection of the common bile duct with subsequent biliodigestive anastomosis should be performed to achieve margin negativity. However, routine bile duct resection during secondary oncologic resection is not recommended due to the increased morbidity and lack of a survival benefit [25,45].

Traditionally, secondary oncologic resections of incidental GBC and primary resection of non-incidental, advanced GBC have been carried out by an open approach. However, advances in minimally invasive surgery enabled surgeons worldwide to perform such procedures without laparotomy [50]. In particular the increasing availability of robotic surgery allow even extended resections, e.g., including biliary or vascular resection and reconstruction, with the same approach as in open surgery [51]. Furthermore, modern techniques such as intraoperative indocyanine green fluorescence can be used to clarify vascular or biliary anatomy, particularly in secondary cases with (post)inflammatory alterations at the hepatouodenal ligament [52]. Although current studies suggest the safety and oncologic adequacy for minimally invasive procedures for GBC, these findings still need to be corroborated in prospective large-scale studies.

In summary, in the absence of high-quality evidence regarding the optimal extent of surgery for GBC, an individual approach respecting the above-mentioned principles to achieve margin-negative resection should be tailored for each patient. This approach should take radiologic findings, the primary operative and histological reports (in incidental GBC), and also conditional factors of the individual patient into account.

### 2.3. Adjuvant Therapy

After curative resection, one-third (33.3%) of GBC patients develop recurrent disease at a median follow-up of 15.1 months [53]. As risk factors for recurrence, operative jaundice, major hepatectomy, T-category 3/4, N-category 1/2, tumor size, poorly differentiated tumor, lymphovascular invasion, and R1 margin status could be identified [53,54]. Horgan et al. have demonstrated in their metanalysis of 6712 patients with BTC that underwent resection between 1960 and 2010 that patients with GBC benefit from adjuvant therapy, especially those with node-positive disease and R1-resection [54]. These results could be replicated in newer studies. Ma et al. conducted a pooled meta-analysis involving 3191 patients with GBC showing a significant improvement of OS in patients with adjuvant chemotherapy after resection, especially in patients with R1-resection, node-positive disease, and stage III/IV disease [55]. It is important to know that Mantripragada et al. have suggested that only a high-risk subset of patients harboring the above-listed risk factors might benefit from adjuvant treatment based on their retrospective analysis of locally advanced GBC patients that received adjuvant treatment after resection [56].

At the present moment, actual guidelines including the National Comprehensive Cancer Network (NCCN), European Society for Medical Oncology (ESMO), British Society of Gastroenterology (BSG), and the International Liver Cancer Association (ILCA) differ in their recommendations concerning the adjuvant treatment of BTC, and a differentiation between GBC and other BTC forms is lacking [57]. The most important data for adjuvant therapy in BTC derives from the phase III BILCAP trial, which investigated prospectively the effect of adjuvant capecitabin in 447 patients with BTC after resection containing 79 patients (18%) with muscle-invasive GBC [58]. Patients in the capecitabine arm showed improved median survival compared with those in the observation arm (51.1 vs. 36.4 months, *p* = 0.028) in the per-protocol analysis but not in the intent-to-treat analysis. Considering adverse events, it must be noted that, out of 213 patients who received at least one cycle of capecitabine, 94 (44%) had at least one grade 3 toxicity (43 patients with hand–foot syndrome, sixteen patients with diarrhea, sixteen patients with fatigue, and nineteen other adverse events). Despite several limitations of this study and lacking subgroup analysis concerning GBC, adjuvant therapy with capecitabine after curative resection for GBC can be considered the current standard of care.

Besides the BILCAP trial, several other phase III trials have investigated the effect of different adjuvant approaches for the therapy of resected BTC with partly contradictory results.

The PRODIGE-12/ACCORD-18 phase III trial has compared the effect of adjuvant chemotherapy with gemcitabine and oxaliplatin (GEMOX) compared to an observational strategy after curative resection for BTC in 196 patients including 38 GBC (19%) patients. In opposition to the results of the BILCAP trial, the authors could not observe any advantage of the study arm considering the disease recurrence rate, relapse-free survival (RFS), and OS. Moreover, a subgroup analysis of patients with GBC showed a significantly worse RFS and OS in the study arm compared to the observation arm. In their discussion section, the authors suggest that the higher sample size leading to a high statistical power as well as the higher amount of high-risk BTC patients with early recurrence (R1-resection status and node-positive disease) might explain the different results compared to the BILCAP trial.

Another clinical phase III trial that has investigated the role of adjuvant chemotherapy in patients with BTC was the Japanese BCAT trial, which recruited 226 patients with BTC (without defining a GBC subgroup) for the evaluation of adjuvant gemcitabine therapy [59]. Disappointingly, the results did not show a difference in RFS, OS, or disease recurrence between the gemcitabine arm and the observation arm, even under consideration of a high-risk subgroup analysis (R1-resected patients, node-positive patients). Similar to the PRODIGE-12/ACCORD-18 trial, the study results have been limited due to an insufficient sample size and a smaller subset of high-risk BTC patients. Furthermore, the results are moreover limited by a lack of GBC patients.

Considering the strong limitations, the value of a clinical phase III trial by Takada et al. that has investigated the effect of an adjuvant chemotherapy with mitomycin c and 5-FU has been discussed [60]. In their study, a heterogenous study collective of 508 patients with pancreatic and BTC including GBC (*n* = 140) has been divided in a study arm that received adjuvant chemotherapy with gemcitabine and mitomycin c and a control group with an observation strategy after curative resection between 1986 and 1992. In the subgroup analysis of GBC patients, the 5-year OS was significantly higher in the study arm (26.0% vs. 14.4%; *p* = 0.0367) accompanied by a significantly improved disease-free survival (20.3% vs. 11.6%; *p* = 0.0210).

Another approach is the application of a combined adjuvant radio-chemotherapy, since GBC disease recurrence is often characterized by local disease relapse [53], and retrospective analysis have shown a benefit from radiotherapy in resected GBC [38,61]. In the single-arm SWOG S0809 phase II trial, Ben-Josef et al. investigated the effect of adjuvant gemcitabine and capecitabine, followed by radio-chemotherapy with capecitabine and radiotherapy with 45 Gy to regional lymph nodes and 54 to 59.4 Gy to preoperative tumor bed in 79 patients, including 25 GBC patients [62]. Considering the lack of a control arm, the results showed a promising OS and DFS without a significant difference between GBC and other BTC. Overall, there was a 2-year OS of 65% with a median follow-up of 35 months and a 2-year DFS of 52%. These results must be further replicated in a prospective phase III trial. The Chinese FDRT-PG001 phase III trial is therefore recruiting 140 patients with GBC and extrahepatic BTC after resection to compare the effect of a combined concurrent radio-chemotherapy (gemcitabine and capecitabine, followed by capecitabine and radiotherapy) with chemotherapy alone (gemcitabine and capecitabine) (NCT02798510). Unfortunately, since this study has not been updated since June 2016, it is unlikely that this trial will support further evidence for the use of radio-chemotherapy in GBC.

At the moment, the ongoing ACTICCA-1 is expected to supply further evidence for the adjuvant treatment of GBC. In their multicenter phase III trial, the authors aim to compare a combined adjuvant chemotherapy regimen of gemcitabine and cisplatin based on the results of the ABC-02 trial [63] with a control arm that receives adjuvant chemotherapy with capecitabin [64]. With an aspiredenrollment of 781 participants with resected BTC including GBC, results are estimated in April 2024.

Besides the ACTICCA-1 trial, other adjuvant chemotherapy regimens are investigated in current phase III trials. The Japanese ASCOT-trial has recruited 440 patients with BTC to compare the effect of adjuvant S-1 therapy with an observational strategy [65]. In their data presentation at the ASCO Gastrointestinal Cancers Symposium 2022, the authors could demonstrate that an adjuvant therapy with S-1 was significantly superior to a observational strategy in terms of OS and PFS for all subgroups that were evaluated (ECOG performance status, age, cancer type, cancer stage, R factor, and serum CA19-9) with tolerable adverse events. However, it must be pointed out that the results might not be applicable to non-Asiatic patients since the tumor biology of GBC has major differences between different ethnic origins (see above).

The Chinese AdBTC-1 phase III trial is currently recruiting an aspired number of 460 patients to investigate the effect of gemcitabine and capecitabine compared to capecitabine alone in resected BTC patients (NCT03779035). Unfortunately, this trial has not been updated since 2018; therefore, further evidence is unlikely.

In conclusion, patients with GBC after curative resection should be considered for adjuvant chemotherapy with capecitabine based on the results of the BILCAP trial, especially in case of risk factors for disease recurrence (R1, N+, G3/4). Nevertheless, it must be admitted that evidence for adjuvant therapy after resection in patients with BTC and even more for patients with GBC remains poor. These limitations have been further highlighted by a systematic review of the Cochrane Database which concluded that there was a lack of evidence for the adjuvant treatment of resected BTC patients and moreover possible harm due to increased adverse events [66].

### 2.4. Palliative Therapy

Since GBC is often diagnosed in an advanced, non-curative stage and OS is poor with approximately 4.5 months under the best supportive care therapy [67,68], effective chemotherapy regimens have been established to improve prognosis. Nevertheless, current evidence is mostly established for BTC in general, and specific approaches for GBC are urgently required since it is estimated that GBC differs from other BTC forms in term of therapy response to different cytostatic agents [69].

For first-line therapy, the results of the ABC-02 study and the Japanese BT-22 study led to the establishment of a combined chemotherapy with gemcitabine (1000 mg/m^2^, day 1 and 8 every three weeks) and cisplatin (25 mg/m^2^, day 1 and 8 every three weeks). The ABC-02 phase III study prospectively evaluated the effect of gemcitabine and cisplatin compared to gemcitabine alone in 410 patients with BTC [63]. Overall, patients who received gemcitabine and cisplatin could significantly benefit from the combined chemotherapy regimen with an OS of 11.7 months compared to 8.1 months in the gemcitabine arm and a median PFS of 8.0 months compared to 5.0 months. Additionally, no significant difference was observed regarding the occurrence of adverse events in both groups. These results were consistent in the subgroup analysis of 149 patients with GBC (36.3%) that were part of the study collective.

In the BT-22 phase 3 trial, a similar study was performed, including 83 patients with BTC with 32 patients (39.0%) with GBC [70]. Like in the ABC-02 trial, a significant benefit was found for patients in the study arm with an OS of 11.2 months compared to 7.7 months and a PFS of 5.8 months compared to 3.7 months. In the subgroup analysis of GBC patients, a slightly poorer OS of 9.1 months compared to 6.7 months was observed.

A meta-analysis of both studies could verify these results including patients with GBC but showed no benefit in OS for patients with an ECOG performance state >2 [71]. Therefore, the authors suggest considering a monotherapy with gemcitabine for patients with poor ECOG performance state. In conclusion, the results of the ABC-02 trial and the BT-22 trial form the basis for the recommendation of a combined chemotherapy regimen with gemcitabine and cisplatin as the first-line therapy for patients with primary non-resectable BTC including GBC. For patients with a poor ECOG performance state, gemcitabine monotherapy can be considered and for patients with advanced chronic kidney failure, cisplatin should be replaced by oxaliplatin [30,55,57,72].

After the failure of gemcitabine and cisplatin, data for second-line chemotherapy in a palliative setting for BTC and moreover for GBC is strongly limited.

The ABC-06 phase III trial compared the effect of FOLFOX (5-FU and oxaliplatin) compared to active symptom control alone in 162 patients with BTC (34 = 21% GBC) after the failure of gemcitabine/cisplatin [73]. Overall, a benefit in terms of OS (6.2 months vs. 5.3 months) could be observed for the study arm and adverse events in both groups remained similar. In the subgroup analysis for GBC patients (*n* = 34), a difference in OS of 5.1 months vs. 4.6 months and a 12-month OS rate of 35.3% vs. 5.9% could be demonstrated with a disease control rate of 47% in GBC patients.

As another option, the GB-SELECT phase II trial evaluated the effect of irinotecan monotherapy compared to capecitabin and irinotecan (CAPIRI) in 98 GBC patients after disease progression with gemcitabine-based first-line treatment [74]. The authors could show that no difference in terms of OS and PFS was observed between both study arms, but adverse events and the need for dose modification was higher in the CAPIRI arm. Overall, a median OS of 6.28 months and a 6-month OS of 54.2% could be demonstrated in the irinotecan arm, which is comparable to the FOLFOX regimen despite differences in terms of study conduct. A phase II clinical trial by Choi et al. compared the effect of second-line treatment with 5-FU and irinotecan (FOLFIRI) to the established FOLFOX regimen in 118 patients with BTC including 35 GBC patients (29.7%) [75]. The authors could demonstrate a similar effect in terms of OS (6.3 months in the FOLFIRI arm vs. 5.7 months in the FOLFOX arm), PFS (2.1 months vs. 2.8 months), and ORR (4.9% vs. 5.9%) between both study arms, with a significant higher occurrence of specific adverse events (thrombocytopenia, peripheral neuropathy, vomiting, and cholangitis) in the FOLFOX arm. Therefore, the authors suggest that FOLFIRI might be a therapeutic option for patients who suffer from adverse events under FOLFOX treatment. As a limitation, it has to be admitted that, in a pooled meta-analysis, only 43.9% of GBC patients were eligible to receive second-line therapy after the failure of first-line therapy [76]. A summary of other clinical trials that have investigated the effect of different chemotherapy regimen is provided in Table 2.

Besides above-named chemotherapy regimen, current clinical trials have evaluated other cytostatic agents in patients with advanced-stage BTC, but data quality remains poor, and no phase III study has been conducted to compare these approaches with established therapy regimens. In a systematic review and meta-analysis, Azizi and colleagues investigated the effect of different cytostatic chemotherapy regimens specifically for patients with advanced GBC [76]. The authors could include 58 studies with 1986 GBC patients that were conducted until March 2019. Overall, an ORR of 23.2% with an OS of 4.8 months and a PFS of 8.3 months could be demonstrated. The pooled analysis suggested a benefit for chemotherapy regimens containing three or four cytostatic agents, with a pooled ORR of 35.8%, an OS of 9.9 months, and a PFS of 5.9 months. Moreover, GBC patients with platinum-based chemotherapy regimens had a benefit in OS, while gemcitabine-based therapy did not show a similar effect. In total, GBC patients showed a higher ORR than non-GBC patients (odds ratio 0.65; 95%CI 0.50–0.84), but OS and PFS could not be reliably compared based on the included study data.

Based on these results, a combined chemotherapy regimen with three or four cytostatic agents might improve prognosis for patients with GBC in a palliative setting and good ECOG state.

In a single-arm phase II clinical trial, Shroff et al. administered combined chemotherapy with nanoparticle-albumin-bound (nab-) paclitaxel, gemcitabine, and cisplatin to a study collective of 62 patients with advanced BTC including 13 GBC patients (22%) [84]. Overall, a median OS of 19.2 months and a median PFS of 11.8 months with a disease control rate of 84% could be achieved. Moreover, no differences between ORR and OS between different tumor sites could be observed. It should be noted that thirty-three patients (58%) suffered from an adverse event of grade 3 or higher, which might limit the group of patients that could tolerate this regimen. Nevertheless, these promising results are investigated in the current Southwest Oncology Group 1815 phase III trial (NCT03768414) to compare this regimen to the current standard therapy with gemcitabine and cisplatin to provide further evidence.

As another treatment option, a phase II trial has investigated the effect of a combined regimen containing 5-FU, leucovorin, oxaliplatin, and irinotecan (mFOLFIRINOX) in 29 patients with unresectable GBC [85]. Despite the high frequency of adverse events of grade 3 or higher in twenty-three patients (79.3%), the study showed encouraging results with an ORR of 48.3% and a disease control rate of 79% leading to an OS of 13.8 months and a PFS of 8.4 months in patients with a response to mFOLFIRINOX therapy. At the moment, the PRODIGE38-AMEBICA phase III study is randomizing 188 patients with advanced-stage BTC from a previous phase II study [86] to compare the effect of mFOLFIRINOX to gemcitabine and cisplatin [87]. In the previous phase II study, the authors recruited 191 patients that were randomly assigned to a mFOLFIRINOX group and a gemcitabine/cisplatin group. Disappointingly, the triplet regimen with mFOLFIRINOX did improve PFS or median OS. Moreover, the doublet group had a slightly better median PFS (7.4 months vs. 6.2 months) and median OS (13.8 months vs. 11.7 months) [86].

Since GBC often occurs in elderly patients who might not be eligible for aggressive chemotherapy regimens in a non-curative setting, palliative radiation therapy might improve local disease control while causing fewer adverse events. Unfortunately, data on the effect of definitive radiation in unresectable GBC patients are lacking, despite the progress in new radiation techniques. As reported above, Verma et al. have suggested that patients with locally advanced GBC benefit from combined radio chemotherapy compared to a solitary chemotherapy regimen, based on their retrospective analysis of 1199 patients with locally advanced, non-metastatic gallbladder cancer from the National Cancer Database [38]. Additionally, Pollom et al. investigated the effect of radiation in a geriatric cohort of 2343 patients > 70 years from the SEER–Medicare database, including 50 GBC patients who received radiation therapy [88]. The authors proposed that patients who received chemotherapy benefit from additional radiation in terms of improved survival with an adjusted hazard ratio of 0.82 (95% 0.70–0.97, *p* = 0.02). The primary tumor site of BTC in their analysis had no effect on these results. In a retrospective single-center analysis of forty-five patients with locally advanced GBC patients receiving chemotherapy alone vs. combined radio-chemotherapy, Sinha et al. confirmed previous findings. In their analysis, a significant benefit for patients receiving radiotherapy was observed in terms of 2-year PFS (18.6% vs. 0%, *p* = 0.0001) and OS (37.3% vs. 5%, *p* = 0.0001). Importantly, adverse events related to radiation therapy were rare. With an ongoing improvement in radiation techniques and imaging technology, radiation therapy might change local disease control in unresectable GBC. As an example, Makita et al. evaluated the effect of additional proton beam therapy in unresectable BTC patients (3 GBC), leading to a 1-year OS of 49.0% and a 1-year PFS of 29.5% with a 1-year local disease control rate of 67.7%. These results were found to be independent from the chemotherapy regimen the included patients received, but patients with a tumor size < 5 cm and an ECOG > 1 were shown to significantly benefit more from a proton beam therapy. Despite these encouraging results, further prospective studies are mandatory to provide more evidence, with a special focus on elderly patients who are not eligible for an aggressive chemotherapy regimen.

Beyond radiation therapy, other approaches for local disease control for BTC are investigated in several studies, but these studies do not specifically address a GBC subgroup, and a detailed description is beyond the scope of this review. Furthermore, endoscopic techniques for symptom control and biliary drainage in patients with GBC are described in a current review by Schepis et al. and will not be discussed in this article [89].

In conclusion, differentiated therapy for patients with non-resectable GBC is mandatory. Nevertheless, a combined chemotherapy regimen of gemcitabine and cisplatin as first-line treatment can be considered the standard of care in line with other forms of BTC due to a lack of data on GBC-specific therapies. For second-line treatment and radiation therapy, data for BTC and moreover for GBC is poor, and the only recommended therapeutic option after the failure of gemcitabine and cisplatin is a chemotherapy with FOLFOX. Further studies might establish a more aggressive chemotherapy regimen based on three to four substances that might improve the prognosis of GBC.

### 2.5. Targeted Therapy and Future Perspectives

As GBC is characterized by unique molecular alterations, current research has focused on the development of new targeted therapy strategies to improve OS and to reduce the number of adverse events caused by cytostatic therapy. An overview of therapeutic targets and agents is provided in Table 3.

#### 2.5.1. HER2/Neu (ErbB2)

HER2 (synonym: ErbB2) is a cell-surface receptor with a transmembrane tyrosine kinase domain that plays a crucial role in tumor biology through the downstream activation of the PI3K/Akt pathway (call polarity, cell adhesion, cell cycle) and the MAPK pathway (mitosis) in multiple cancers [90,91]. In GBC, HER2/neu overexpression is more common than in other forms of BTC, with a frequency of 9.8 up to 27.3% [92,93,94]. In a transgenic mice model, Kiguchi et al. have demonstrated that HER2/neu amplification in the basal layer of the biliary tract epithelium led to the development of gallbladder cancer within 3 months [95]. In general, the amplification of HER2/neu was found to be associated with a worse prognosis in patients with GBC [94].

The MyPathway multicenter, open-label, phase 2a, multiple basket trial is evaluating the effect of a combined regimen of pertuzumab and trastuzumab in patients with BTC and Her2 amplification or overexpression [96]. In a preliminary analysis, a promising ORR of 23% in 39 recruited patients was observed, leading to a PR in nine patients.

Additional evidence for the use of Her2/neu-directed therapy in GBC patients is provided by the results of the open-label, single-arm, multi-cohort phase 2 SUMMIT basket trial that evaluated the effect of neratinib, an irreversible pan-HER oral tyrosine kinase inhibitor, in patients with solid tumors harboring oncogenetic HER2 somatic mutations. In a preprint, the treatment results of the biliary tract cancer cohort (*n* = 25) including 10 patients with GBC (40%) were published by Harding and colleagues [97]. Among ten pretreated GBC patients harboring Her2 mutation, three patients had a confirmed PR after receiving neratinib, leading to a PFS of 3.7 months and an OS of 9.8 months. Within the whole BTC cohort, a ORR of 16% and a clinical benefit rate (confirmed complete or partial response or stable disease for at least 16 weeks) of 28% was observed. In general, the authors conclude that especially BTC patients with GBC might benefit from a combined chemotherapy regimen including neratinib for Her2 mutated tumors.

In a prospective pilot study of trastuzumab in combination with gemcitabine and cisplatin in patients with BTC, Jeong et al. demonstrated preliminary feasibility regarding adverse events, but data on therapeutic response were limited due to the fact that only four patients with HER2/neu amplification and only one patient with GBC were included [98]. Interestingly, even GBC patients without Her2/neu amplification might benefit from ErbB2-directed therapy after pretreatment with 5-FU or gemcitabine. Wang et al. demonstrated that HER2-negative GBC cells that were treated with gemcitabine/5-FU developed the strong expression of HER2/neu that led to a strong response to trastuzumab therapy. Therefore, a sequential therapeutic strategy might sensitize HER2-negative GBC patients for a ErbB2-directed therapy [99].

Another promising approach to target Her2 mutated BTC is the use of trastuzumab deruxtecan (DS-8201), an antibody–drug conjugate composed of a humanized monoclonal anti-HER2 antibody, a cleavable tetrapeptide-based linker, and a potent topoisomerase I inhibitor (as a cytotoxic agent). The phase II HERB trial (JMA-IIA00423) aims to evaluate the effect of trastuzumab deruxtecan in 32 patients with advanced Her2-expressing BTC [100]. In a preliminary data analysis presented at the ASCO annual meeting 2022, the authors could demonstrate an ORR of 36.4% and a DCR of 81.8%, leading to an PFS of 4.4 months and a median OS of 7.1 months in 22 BTC patients including 11 GBC patients (50%) [101]. For further evidence, the subgroup analysis of GBC patients is highly anticipated.

At the moment, several clinical trials are recruiting patients with BTC and GBC to investigate the effect of ErbB2-directed therapy alone or combined with different chemotherapy regimens, e.g., trastuzumab and FOLFOX (NCT04722133; completion in March 2023), afatinib and capecitabine (NCT02451553; completion in September 2022), and a combined anti-ErbB2 regimen of tucatinib and trastuzumab (NCT04579380).

#### 2.5.2. VEGF/VEGFR and Antiangiogenetic Therapy

Since tumor angiogenesis has been identified as a key factor for cancer progression and metastasis, therapeutic approaches that target molecular pathways for tumor vascularization have been established in several cancer forms, e.g., colorectal cancer [102] and breast cancer [103]. For GBC, VEGF-A overexpression that mediates angiogenesis was found to be an independent prognostic factor for worse survival [104]. In accordance with these results, Xu et al. demonstrated the crucial role of VEGF overexpression for angiogenesis, cell proliferation, and metastasis in GBC cells [105]. Up to 48% of GBC patients show a strong expression of VEGF-A that might lead to a benefit from antiangiogenetic agents [106]. Several studies have investigated the effect of therapeutic agents that address tumor angiogenesis via the VEGF/VEGFR pathway. In a clinical phase II trial, Zhu et al. investigated the effect of a combined regimen of bevacizumab, gemcitabine, and oxaliplatin in 35 patients with advanced BTC [107]. A response rate of 40% with a PFS of 7 months and an OS of 12.7 months could be achieved. In a single-arm phase II trial of 43 patients with chemotherapy-refractory, unresectable BTC, therapy with regorafenib could achive a median PFS of 15.6 weeks and a median OS of 31.8 weeks with a disease control rate of 56% [108]. Furthermore, a combined regimen of erlotinib and regorafenib in 53 BTC patients showed a median OS of 9.9 months and a time to progression (TTP) of 4.4 months [109]. In contrast to these findings, other clinical trials did not find a benefit in BTC patients who were treated with antiangiogenetic agents. In a multicenter phase II trial that investigated the effect of gemcitabine, capecitabin, and bevacizumab in 50 patients with BTC (22% GBC), the addition of bevacizumab did not improve the outcome in comparison to historical cohorts that were treated with gemcitabine and capecitabin alone [110]. Other studies have shown disappointing results for the use of sorafenib [111], sunitinib [112], vandetanib [113], and ramucirumab [114]. Furthermore, all of the above-described studies did not define a subgroup with high VEFG expression nor were they conducted in a cohort with GBC patients only. Since the molecular effect of antiangiogenetic substances differs between strictly targeted substances and multitargeted substances, further studies are needed to identify suitable agents and a subgroup of GBC patients with distinctive molecular alterations that might benefit from an antiangiogenetic therapy.

#### 2.5.3. EGFR (ErbB1; HER1)

EGFR (synonym ErbB1, HER1) plays a crucial role in cancer proliferation, cell mortality, angiogenesis, cell adhesion, and metastasis through multiple downstream pathways (ERK/MAPK, PI3K-AKT, SRC, PLC-γ1-PKC, JNK, JAK-STAT) [115,116,117]. The overexpression of EGFR in GBC is common, with a frequency of 44–77% [118,119,120], and is associated with a worse prognosis [121]. At the moment, data on the effect of anti-EGFR therapy in GBC patients is disappointing. In a clinical phase II trial in 42 patients with BTC including 16 GBC patients, therapy with erlotinib could only achieve an overall response rate of 8%, with no association between the EGFR level and clinical outcome [122]. In a SWOG phase II trial, the combination of sorafenib and erlotinib in 14 patients with GBC and 20 patients with BTC showed an overall response rate of 6% with a PFS of 2 months in eligible patients [123]. In a meta-analysis by Cai et al., the authors evaluated the effect of EGFR-targeted therapy with GEMOX in 612 BTC patients [124]. Despite a significant benefit in terms of PFS and ORR, no benefit concerning OS could be demonstrated. Moreover, a subgroup analysis showed that the significant benefit in terms of ORR and PFS could not be observed in the GBC subgroup. The authors conclude that only patients with ductal cholangiocarcinoma might benefit from an anti-EGFR therapy.

#### 2.5.4. MAPK (RAS/RAF/MEK/ERK) Pathway

The MAPK signaling pathway is known to be a key factor in the regulation of cell proliferation and apoptosis and is frequently dysfunctional in cancer cells [125]. The intracellular or membrane-receptor-mediated activation of RAS protein leads to the downstream activation of RAF kinase, MEK kinase, and finally ERK phosphorylation, which ultimately activates the transcription factors that regulate gene expression [126]. Since KRAS point mutation is present in up to 41% of GBC and BRAF gene amplification is present in 5% of GBC patients, with huge geographical differences, therapeutic agents that interact with the MAPK signaling pathway are tested in clinical trials [127]. In a single-arm phase IIa clinical trial, Ikeda et al. investigated the effect of trametinib, a MEK-inhibitor, in 20 patients with gemcitabine-refractory BTC (*n* = 8 GBC) [128]. Treated with 2 mg trametinib as a single dose daily, 65% of all patients had a stable disease, and a PFS of 10.6 months could be achieved, leading to a 1-year OS of 20%. Contrary to these findings, the SWOG S1310 study was closed early due to a superior OS (6.6 months vs. 4.3 months) and median PFS (3.3 months vs. 1.4 months) for 44 BTC patients (32% GBC) treated with 5-FU or capecitabine therapy compared to trametinib [129]. For further evidence on the effect of MEK inhibitor therapy in GBC, the phase I ABC-04 trial presented a therapy regimen based on the MEK inhibitor selumetinib combined with gemcitabine and cisplatin [130]. In eight BTC patients, a median PFS of 6.4 months could be observed. Furthermore, a multicentric phase II study by Bekaii-Saab et al. on the effect of selumetinib in 28 patients with BTC could show a PFS of 3.7 months and a median OS of 9.8 months with mild toxicity [131]. Disappointingly, the well-targetable BRAF V600e mutation seems to be rare in BTC and restricted to intrahepatic BTC [132].

#### 2.5.5. PI3K/AKT/mTOR Pathway

One of the most important pathways for carcinogenesis and metastasis is the PI3K/AKT/mTOR pathway, which is crucial for the development of GBC [133,134,135]. By transmembrane or intracellular activation, a phosphorylation cascade starting from the activation of PI3K is initiated that includes the activation of AKT, PTEN, and mTOR [136]. Besides cell-based models, first clinical trials are evaluating substances that target the PI3K/AKT/mTOR pathway. In a phase I trial, a chemotherapy regimen with everolimus, a mTOR inhibitor, and gemcitabine and cisplatin achieved a stable disease in six of ten BTC patients, suggesting a possible benefit [137]. A phase II clinical trial that investigated the effect of the AKT-inhibitor MK-2206 in eight therapy-refractory BTC patients was closed early due to a lack of therapeutic response [138]. As an obstacle in other trials that investigate the effect of a targeted therapy, the authors suggest that the lack of reliable biomarkers and patient selection led to their results. Up to now, therapeutic agents interfering with the Pi3K/AKT/mTOR pathway that have been evaluated in larger clinical trials regarding their effect on GBC are still lacking.

#### 2.5.6. PD-1/PD-L1, MSI-High, High TMB and ICI Therapy

In recent years, the discovery and clinical implementation of therapeutic agents that stimulate immune-mediated antitumoral effects have changed cancer treatment. By expressing PD-L1 as a surface protein, tumor cells interact with the PD-1 receptor that is expressed on T cells and thus suppress an antitumoral T-cell-mediated immune reaction [139]. Importantly, HER2/neu mutation and the upregulation of the PI3K/Akt signaling pathway (see above) in tumor cells are causing the upregulation of PD-1 expression [140]. Furthermore, Gong et al. showed that the upregulation of PD-1 mediates tumor resistance against cytostatic chemotherapy with gemcitabine and oxaliplatin, and therapeutic interventions that target PD-1/PD-L1 interaction might contribute to overcome drug resistance [141]. As a predictor for therapy response to immune checkpoint inhibitors (ICI) that block PD-1/PD-L1 interaction, a high tumor mutational burden (TMB), a high level of microsatellite instability (MSI-high), and a high expression of PD-L1 in tumor cells have been identified [127]. In a sample of 428 GBC patients, Weinberg et al. found a high TMB in 1.0%, MSI-high in 5.8%, and PD-L1 overexpression in 8% [142]. In conclusion, 12% of GBC patients might benefit from ICI therapy due to specific genetic alterations. Therefore, several studies have investigated the effect of ICI therapy in GBC with encouraging results. After several promising phase I studies that have shown a favorable efficacy of ICI therapy [143,144,145,146], further phase II clinical trials have already confirmed these results. The KEYNOTE-158 phase II study has evaluated the effect of pembrolizumab in 104 patients with advanced BTC based on the previous phase I KEYNOTE-028 trial [143]. The authors could demonstrate an ORR of 5.8% with an OS of 7.4 months and a PFS of 1.8 months. Disappointingly, the authors did not define a GBC subgroup that limits the transferability of the results. Another single-arm phase II clinical trial evaluated the effect of nivolumab in 54 patients with pretreated BTC (31% GBC; *n* = 17) [147]. In the intention-to-treat analysis, a PFS of 3.68 months and an OS of 14.24 months was demonstrated. In the GBC subgroup, an ORR of 15% was observed. Interestingly, PD-L1 status and MSI-high were not found to predict therapy response. The most important study that might change the first-line treatment of patients with advanced BTC is the double-blind phase III TOPAZ-1 study that has investigated the effect of a combined regimen of gemcitabine, cisplatin and durvalumab compared to gemcitabine and cisplatin [148]. In an interim analysis in August 2021, 685 patients could be assessed who were randomized 1:1. Patients who received durvalumab in addition to gemcitabine and cisplatin had a significantly improved median OS (12.8 vs. 11.5 months), 24-months OS rate (24.9% vs. 10.4%), and PFS (7.2 vs. 5.7 months) with comparable adverse events. These findings have been confirmed in the final publication in June 2022, additionally containing a subgroup analysis of the included 171 GBC patients (25.0%) [149]. In the subgroup analysis, no differences in terms of PFS and OS was observed regarding the primary tumor site, PD-L1 expression level, or sex or ethnic background.

#### 2.5.7. DNA Damage Repair (DDR) Deficiency

Alterations in the DNA damage repair pathway (mutation of ATM, ATR, BRCA1, BRCA2, FANCA, FANCD2, MLH1, MSH2, MSH6, PALB2, POLD1, POLE, PRKDC, RAD50, SLX4BAP1, CDK12, MLL3, TP53, BLM [150]) are a field of research that has led to new therapeutic agents that specifically address these molecular changes, as well as evidence about specific tumor vulnerability for platinum-based chemotherapy regimens [151]. Kumari et al. evaluated the prevalence of DDR deficiency in 111 GBC samples. In a preprint, the authors describe a prevalence of 27.6% for DDR deficiency (MLH-1, MSH-6, MSH-2, and PMS-2) [152]. Other studies have found a lower prevalence of the DDR deficiency of GBC. Javle et al. identified 7.8% of 623 GBC tissue samples as having the BRCA2 mutation or ATM mutation [153]. To target these alterations, PARP inhibitors have been developed and showed promising results in several cancer forms, e.g., in pancreatic ductal adenocarcinoma [154]. At the moment, only a case report of a patient suffering from ATM-mutated GBC and treated with olaparib with a PFS of 13 months is available [155]. Therefore, several prospective phase II clinical trials are ongoing that evaluate the effect of olaparib (NCT04042831), rucaparib, and nivolumab (NCT03639935); nal-irinotecan/5FU and rucaparib (NCT03337087); and rucaparib alone (NCT04171700) for BTC [156].

#### 2.5.8. Other Molecular Alterations and Future Perspectives of Targeted Therapy in GBC

In GBC, TP53, which functions as a tumor suppressor gene, is alternated in approximately 53% of all patients [24], but efforts to develop a promising therapeutic agent that specifically targets this alteration failed. Like TP53, the CDKN2A/B mutation was found to be mutated in 5.9% of GBC patients with a key role in pathogenesis, but no therapeutic agent is currently available to target this alteration [127]. Disappointingly, FGFR2 translocation, which can be targeted with FGFR-inhibitors (pemigatinib, infigratinib, derazantinib, erdafitinib), as well as IDH1/2 mutation (targetable with IDH-inhibitors, e.g., ivosidenib) and NTRK fusion (targetable with entrectinib and larotrectinib), are rare in GBC (FGFR2-translocation 3%, IDH1/2 mutation 2%, NTRK fusion 4%) [24]. Furthermore, data on the efficacy of the above-named targeted therapies specifically for GBC is overall poor [24]. A promising target might be mutations of the C-mesenchymal-epithelial transition factor (c-MET) that is present in up to 74% of all GBC [157]. At the moment, several therapeutic agents including small molecules targeting MET receptors (e.g., crizotinib, tivantinib, savolitinib, tepotinib, cabozantinib, and foretinib), MET receptor monoclonal antibodies (e.g., onartuzumab), and antibodies against its ligand HGF (e.g., ficlatuzumab and rilotumumab) are available and showed promising results in other C-MET-mutated solid cancers [127]. In a phase I clinical trial, a therapy regimen containing the C-met inhibitor merestinib and gemcitabine/cisplatin was evaluated in 19 patients, including eight BTC patients who showed a tolerable toxicity and safety [158].

MYC (myelocytomatosis viral oncogene homolog) is a proto-oncogene that promotes cell proliferation by inducing target genes of transcription factors and antagonizing cell-cycle inhibitors [159]. A gain of MYC copies is described in up to 86.7% and MYC mutations in up to 80% of GBC cases, according to Ishak et al. [160], although Kuipers and colleagues report a frequency of MYC mutation between 7–8% in GBC [20]. MYC amplification seems to be related to the coamplification of ERBB2 and EGFR, which can be targeted with specific new approaches (see above) [161]. Unfortunately, developing therapy approaches to target MYC have failed in the past, and new MYC inhibition strategies are far away from clinical implication, although they have been evaluated for the therapy of lung cancer [162].

In conclusion, GBC is characterized by multiple genomic alterations that can be targeted by several new pharmaceutics. In particular, PD-1/PD-L1 inhibitor therapy is estimated to become a new standard of care in combination with the current first-line therapy with gemcitabine and cisplatin. Nevertheless, clinical trials that supply data on the tumor localization of BTC and the lack of predictive biomarkers for targeted therapy impair the clinical feasibility of these therapeutic agents for GBC. With the rising importance of tailored therapy approaches based on molecular profiling and new evidence of molecular differences between different BTC forms, it remains to be hoped that a future therapy regimen might improve the prognosis for GBC patients.

## 3. Conclusions

Gallbladder cancer is the most common malignancy of the biliary tract and is associated with a poor prognosis, as only around 10% of patients are eligible for primary curative surgery. Since previous trials have often evaluated different therapeutic approaches for biliary tract cancer in general without subgroup analysis of GBC, specific therapeutic approaches were lacking. With increasing insight in the unique tumor biology and therapy response of GBC compared to other forms of BTC, current clinical trials have focused on differentiated therapy for different primary tumor sites of BTC including GBC. At the moment, surgical resection is the only curative therapy option, although the appropriate extent of resection has still not been fully investigated for locally advanced GBC. For locally advanced GBC, adjuvant chemotherapy might transform up to one-third of primary non-resectable patients into eligible patients for resection, leading to an improvement in prognosis. After surgery, only a subset of high-risk patients (R1, N+, G3/4) might benefit from adjuvant treatment. In general, data on radiochemotherapy versus chemotherapy alone is still controversial for adjuvant and neoadjuvant treatment. For non-resectable GBC, a palliative chemotherapy regimen of gemcitabine and cisplatin according to the ABC-02 trial has been established as the standard of care, in line with the treatment of other forms of non-resectable BTC. After the failure of gemcitabine and cisplatin, the FOLFOX regimen is the only established treatment option for second-line therapy. The unique tumor biology of GBC harbors the opportunity to improve patients’ prognosis by molecular targeted therapy approaches. The most promising approach is the use of the PD-L1 inhibitor durvalumab combined with gemcitabine and cisplatin that is assumed to become the new standard of care for non-resectable GBC. Other promising approaches include antiangiogenic therapy, Her2/neu-directed therapy, and a combined chemotherapy regimen of 3–4 cytostatic agents for patients with good ECOC performance state.

## Figures and Tables

**Figure 1 cancers-14-05580-f001:**
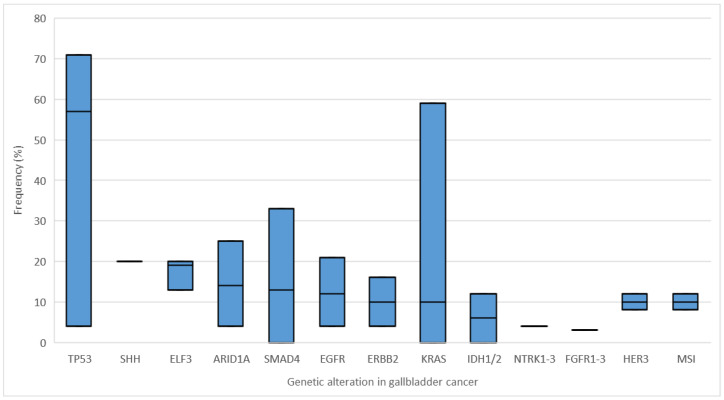
Common mutations in gallbladder cancer with average frequency and range (minimum and maximum) summarized from [20,21,22,23,24]. Average frequency (%) is displayed as a horizontal black line (average frequency without a range given is displayed as a black line only).

**Table 1 cancers-14-05580-t001:** Ongoing clinical trials for the evaluation of neoadjuvant therapy in locally advanced gallbladder cancer. BTC: biliary tract cancer. GBC: gallbladder cancer. OS: overall survival. ORR: overall response rate. RCT: radiochemotherapy. RT: radiotherapy.

NCT Number	Study Phase	Condition	Study Size	Treatment Agent	Primary End Point	Institution	Completition
NCT03673072	III	Incidental GBC and BTC	300	Gemcitabine + cisplatin perioperative vs. adjuvant	OS	Krankenhaus Nordwest, Germany	November 2024
NCT02867865	II/III	GBC	314	Gemcitabine + cisplatin alone vs. RT + gemcitabine + cisplatin	OS	Tata Memorial Hospital, India	August 2022
NCT04308174	II	BTC	45	Gemcitabine + cisplatin vs. gemcitabine + cisplatin + durvalumab	R0 resection rate	Asan Medical Center, Korea	December 2023
NCT04559139	II/III	GBC	186	Gemcitabine + cisplatin perioperative vs. adjuvant	OS	Emory University, Winship Cancer Institute, United States	September 2023
NCT04480190	I	BTC	12	Gemcitabine + cisplatin followed by RCT (5FU + RT)	Therapy completion	University of Cincinnati Medical Center, United States	February 2029

**Table 2 cancers-14-05580-t002:** Clinical trials for the palliative cytostatic chemotherapy of BTC specifically including GBC patients.

NCTN	Phase	Condition	Study Size	Substance	Results	Reference
NCT00660140	II	BTC + GBC	49	gemcitabine + carboplatin	PFS 7.8 months, OS 10.6 months	[77]
Not applicable	II	GBC	20	gemcitabine + carboplatin	ORR 36.7%, PFS 33.8 weeks	[78]
NCT00009893	II	BTC + GBC	42	gemcitabine + 5FU + leucovorin	PFS 4.6 months, OS 9.7 months	[79]
NCT00003276	II	BTC + GBC	39	irinotecan	ORR 8%	[80]
NCT00033540	II	BTC + GBC	57	gemcitabine + capecitabin	ORR 25%, OS 7 months	[81]
NCT00059865	II	BTC + GBC	63	gemcitabine + pemetrexed	No benefit of combined regimen compared to gemcitabine	[82]
NCT00075504	II	BTC + GBC	33	triapine + gemcitabine	ORR 9%; no benefit with triapine	[83]

**Table 3 cancers-14-05580-t003:** Overview of targetable molecular alterations in GBC.

Molecular Alteration	Frequency	Therapeutic Agents
HER2/Neu overexpression/amplification	9.8–27.3%	trastuzumab, lapatinib, neratinib, pertuzumab, afatinib, tucatinib
VEGF high expression	48%	bevacizumanb, sorafenib, sunitinib, ramucirumab, vandetanib
EGFR overexpression	44–77%	erlotinib, cetuximab, panitumumab, gefitinib, afatinib, dacomitinib, osimertinib, olmutinib
MAPK pathway alteration	Up to 45%	trametinib, selumetinib, sorafenib, tipifamib
PI3K/AKT/mTOR pathway alterations	10%	everolimus
MSI-high/high TMB/high PD-L1 expression	12%	pembrolizumab, nivolumab, durvalumab, tremelimumab
DDR deficiency	Up to 27%	olaparib, niraparib, rucaparib
IDH1/IDH2 mutation	2%	ivosidenib
NTRK fusion	4%	entrectinib, larotrectinib
FGFR2 translocation	3%	pemigatinib, infigratinib, derazantinib, erdafitinib
c-MET overexpression	Up to 74%	small molecules targeting MET receptors (e.g., crizotinib, tivantinib, savolitinib, tepotinib, cabozantinib, and foretinib), MET receptor monoclonal antibodies (e.g., onartuzumab), and antibodies against HGF (e.g., ficlatuzumab, rilotumumab)

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
