# Peer review of "Gallbladder Cancer: Current Multimodality Treatment Concepts and Future Directions"

_cancers, 2022, doi:10.3390/cancers14225580_

Round 1
Reviewer 1 Report (Previous Reviewer 1)
The authors have answered the questions raised and adjusted the manuscript accordingly. I think it is a useful addition to the existing literature.
Author Response
Reviewer 1:
Comments and Suggestions for Authors: The authors have answered the questions raised and adjusted the manuscript accordingly. I think it is a useful addition to the existing literature.
Answer: Dear Reviewer 1, thank you very much for your annotations that have improved our manuscript!
Reviewer 2 Report (Previous Reviewer 2)
The manuscript was substantially improved compared with the original article.
However, there are some parts where the authors did not appropriately respond to the reviewer’s comments. Therefore, I think further revision is necessary. The followings are comments to the authors again.
According to the inclusion criteria presented in ClinicalTrials.com, NCT04824742 and NCT03603834 do not include patients with gallbladder cancer.
For NCT04824742, the Condition is ‘resectable cholangiocarcinoma.’
For NCT03603834, the Condition is ‘cholangiocarcinoma’.
Cholangiocarcinoma (or bile duct cancer) refers to cancers originating in the bile duct and do not include gallbladder cancer.
Please see the following websites.
https://www.cancer.gov/types/liver/bile-duct-cancer
https://www.mayoclinic.org/diseases-conditions/cholangiocarcinoma/symptoms-causes/syc-20352408
Therefore, the abovementioned two trials are not ongoing clinical trials for evaluating neoadjuvant therapy in locally advanced gallbladder cancer.
Additionally, NCT04727541 has already been terminated (https://www.clinicaltrials.gov/ct2/show/record/NCT04727541), therefore this trial should not be included in the ongoing clinical trials for evaluating neoadjuvant therapy in locally advanced gallbladder cancer.
Author Response
Reviewer 2:
Comments and Suggestions for Authors:
The manuscript was substantially improved compared with the original article.
However, there are some parts where the authors did not appropriately respond to the reviewer’s comments. Therefore, I think further revision is necessary. The followings are comments to the authors again.
Answer: Dear Reviewer 2, thank you again for your precious annotations concerning our manuscript. We have addressed all your concerns and elaborated our manuscript again based on your comments.
Question: According to the inclusion criteria presented in ClinicalTrials.com, NCT04824742 and NCT03603834 do not include patients with gallbladder cancer.
For NCT04824742, the Condition is ‘resectable cholangiocarcinoma.’
For NCT03603834, the Condition is ‘cholangiocarcinoma’.
Cholangiocarcinoma (or bile duct cancer) refers to cancers originating in the bile duct and do not include gallbladder cancer.
Please see the following websites.
https://www.cancer.gov/types/liver/bile-duct-cancer
https://www.mayoclinic.org/diseases-conditions/cholangiocarcinoma/symptoms-causes/syc20352408
Therefore, the abovementioned two trials are not ongoing clinical trials for evaluating neoadjuvant therapy in locally advanced gallbladder cancer.
Answer: Dear Reviewer 2, thank you for pointing out this critical mistake, we have inspected the above-mentioned trials again and we must acknowledge that these trials did not include GBC patients. Therefore, we have deleted these trials from our manuscript.
Question: Additionally, NCT04727541 has already been terminated
(https://www.clinicaltrials.gov/ct2/show/record/NCT04727541), therefore this trial should not be included in the ongoing clinical trials for evaluating neoadjuvant therapy in locally advanced gallbladder cancer.
Answer: Dear Reviewer 2, thank you again for your detailed analysis of our manuscript. Since we have not found a publication of the results of NCT04727541, we have deleted the study from Table 1 and we have not further discussed this trial in our manuscript.
This manuscript is a resubmission of an earlier submission. The following is a list of the peer review reports and author responses from that submission.
Round 1
Reviewer 1 Report
The authors of this study have established an extensive review on the pathogenesis, multimodal therapy and future directions of gallbladder cancer. The manuscript is in well-written English. It gives a good insight in the current different therapies for gallbladder cancer, the results and limitations of previous clinical trials and an overview over current clinical trials.
There are, however, some major issues to address.
First, it remains unclear what additional value this review provides over the current available literature and reviews. This is missing in the Introduction section.
Second, the 'Pathogenesis' section is not well elaborated. For clarity, it might be better to focus on the different therapies and future perspectives only in this review.
Moreover, there are a few minor issues that need attention:
- Pathogenesis section, paragraph 2.4-2.15 and Figure 1: A systematic review on genetic alterations is required to address the mutation frequencies in gallbladder cancer. Referencing to only one study is insufficient to provide accurate mutation frequencies, as the frequencies are submissive to geographical differences and etiological differences. It would be better to provide mutation ranges from lowest to highest from different reports on each mutation.
- Pathogenesis section, paragraph 2.9: There is detailed information on the MYC gene and the targeted therapies around MYC. The reviewer believes this information should belong to the targeted therapy section.
- Pathogenesis section, paragraph 2.5-2.11: Reference 21 is not reflecting the original study, please adjust this reference.
- Therapy section, first paragraph ('BC.. therapeutics'). It is somewhat unclear why this paragraph handles BC and not GBC, please rewrite this paragraph.
- For the readability of the manuscript, it would be better to divided some long sentences into several sentences (for instance, line 168-173 and 207-213).
- Please rephrase line 717-718 (12.8 vs 11.5 months; 24-months OS 24.9 vs 10.8 months), as it is unclear what is meant by this.
Author Response
Reviewer 1:
The authors of this study have established an extensive review on the pathogenesis, multimodal therapy and future directions of gallbladder cancer. The manuscript is in well-written English. It gives a good insight in the current different therapies for gallbladder cancer, the results and limitations of previous clinical trials and an overview over current clinical trials.
There are, however, some major issues to address.
Dear Reviewer 1, thank you very much for your helpful advice regarding our manuscript. Subsequently, you can find the point by point response.
Question: First, it remains unclear what additional value this review provides over the current available literature and reviews. This is missing in the Introduction section.
Answer: The additional value of this review is to be up to date, including most recent advances in a currently relevantly progressing therapeutic landscape in GBC. Amongst others this includes new innovative therapy approaches (e.g., the TOPAZ I trial) and the role of radiation therapy that have only be discussed in reviews that have focused on particular therapeutical aspects (e.g., only targeted therapy). Moreover we can give a future outlook that is driven by personalization of therapy in GBC patients.
We have highlighted this issue now in our introduction section.
Question: Second, the 'Pathogenesis' section is not well elaborated. For clarity, it might be better to focus on the different therapies and future perspectives only in this review.
Answer: We fully agree with the reviewer, this review should be focused on clinical aspects and only indicate pathological aspects were needed. In line the Pathogenesis section has been integrated into a broader introduction section and the title of this review has been changed into “Gallbladder cancer: Current Multimodality Treatment Concepts Pathogenesis, multimodal therapy, and Future Directions” to reflect the focus on the therapeutical options for GB cancer.
Moreover, there are a few minor issues that need attention:
Question: Pathogenesis section, paragraph 2.4-2.15 and Figure 1: A systematic review on genetic alterations is required to address the mutation frequencies in gallbladder cancer. Referencing to only one study is insufficient to provide accurate mutation frequencies, as the frequencies are submissive to geographical differences and etiological differences. It would be better to provide mutation ranges from lowest to highest from different reports on each mutation.
Answer: The frequency of genetic alterations in the introduction section (former Pathogenesis) has been edited giving a range in addition to each frequency whenever possible (for FGFR13-, NTRK1-3 as well as SHH no range could be found). Furthermore, Figure 1 has been changed into a boxplot graphic that reflects these changes.
Question: Pathogenesis section, paragraph 2.9: There is detailed information on the MYC gene and the targeted therapies around MYC. The reviewer believes this information should belong to the targeted therapy section.
Answer: The MYC section has been integrated into our targeted therapy section.
Question: Pathogenesis section, paragraph 2.5-2.11: Reference 21 is not reflecting the original study, please adjust this reference.
Answer: Reference 21 has been adjusted to the original source by Lamarca et al.
Question: Therapy section, first paragraph ('BC.. therapeutics'). It is somewhat unclear why this paragraph handles BC and not GBC, please rewrite this paragraph.
Answer: The first paragraph of the therapy section, BTC has been changed into GBC which was actually intended
Question: For the readability of the manuscript, it would be better to divided some long sentences into several sentences (for instance, line 168-173 and 207-213).
Answer: The authors have tried to avoid long sentences in terms of a better understanding
Question: Please rephrase line 717-718 (12.8 vs 11.5 months; 24-months OS 24.9 vs 10.8 months), as it is unclear what is meant by this.
Answer: Line 717-718 has been rephrased since there was a mistake in terms of 24-months OS (months were replaced by percent)

Reviewer 2 Report
In this review article, the authors summarized the current evidence on the treatment of gall bladder cancer, including surgery, neoadjuvant, adjuvant, and palliative therapy. The manuscript is well-organized. However, some points need further revision.
The followings are comments to the authors.
On page 2, line 51, ‘oder’ should be ‘or’.
On page 4, line 144, ‘BC’ should be ‘GBC’.
In Table 1, NCT04824742, NCT03603834, and NCT04727541 do not enroll the patients with GBC according to the inclusion criteria presented in ClinicalTrials.com. Therefore, I recommend the authors notify the above information. The sentence ‘Other trials that are currently recruiting participants for the evaluation of neoadjuvant treatment in GBC or BTC including GBC are summarized in Table 1’ may be incorrect.
According to ClinicalTrials.com, NCT02798510, and the NCT03779035, the study status is not updated since 2016 and 2018, respectively. It is beneficial for readers to provide updated information for each study.
The article of reference numbers #50 and #79 are the same.
Regarding the cited article (reference number #102), the study results are already presented in the JCO. I think it is better to discuss the results of the study.
Author Response
Reviewer 2:
In this review article, the authors summarized the current evidence on the treatment of gall bladder cancer, including surgery, neoadjuvant, adjuvant, and palliative therapy. The manuscript is well-organized. However, some points need further revision.
Dear Reviewer 2, thank you very much for your valuable assistance. We have elaborated our manuscript regarding your advice and hope that the revised manuscript addresses all issues.
The followings are comments to the authors.
Question: On page 2, line 51, ‘oder’ should be ‘or’.
Answer: Oder has been changed into or (Former page 2, line 51)
Question: On page 4, line 144, ‘BC’ should be ‘GBC’.
Answer: BC has been changed into GBC (former page 4, line 144)
Question: In Table 1, NCT04824742, NCT03603834, and NCT04727541 do not enroll the patients with GBC according to the inclusion criteria presented in ClinicalTrials.com. Therefore, I recommend the authors notify the above information. The sentence ‘Other trials that are currently recruiting participants for the evaluation of neoadjuvant treatment in GBC or BTC including GBC are summarized in Table 1’ may be incorrect.
Answer: Table 1: NCT04824742, NCT03603834 and NCT 04727541 are recruiting patients with biliary tract cancer, which is mentioned in the section “condition” as BTC. Since few studies have investigated therapy approaches for GBC only, we have included these studies in this table since subgroup analysis of included GBC patients might bring further evidence for GBC in particular.
Question: According to ClinicalTrials.com, NCT02798510, and the NCT03779035, the study status is not updated since 2016 and 2018, respectively. It is beneficial for readers to provide updated information for each study.
Answer: NCT02798510 and NCT03779035 have not been updated since the review of this manuscript. Therefore, we have added a notification about the status of these trials in the relevant section.
Question: The article of reference numbers #50 and #79 are the same.
Answer: Duplication of Ref. 50 and 79 have been corrected.
Question: Regarding the cited article (reference number #102), the study results are already presented in the JCO. I think it is better to discuss the results of the study.
Answer: Ref. 102 (now Ref. 86) has been elaborated with the study results

Reviewer 3 Report
The authors wrote a review on gallbladder cancer: Pathogenesis, multimodal therapy, and future directions. This is a comprehensive review in terms of gallbladder cancer. This is a relatively well written review on the etiology of gallbladder cancer. In particular the pathogenesis and multimodal management through the use of targeted therapy is particularly well written.
The strength of the manuscript is on the pathogenesis as well as the multimodal therapy. Although a minor criticism is that surgical therapy is not extensively covered in management of gallbladder cancer. Surgical treatment remains the only curative therapy for early stage gallbladder cancer. The authors may benefit from expanding on the use of surgical techniques including laparoscopic and robotic surgeries which are arising in frequency.
Author Response
Reviewer 3:
The authors wrote a review on gallbladder cancer: Pathogenesis, multimodal therapy, and future directions. This is a comprehensive review in terms of gallbladder cancer. This is a relatively well written review on the etiology of gallbladder cancer. In particular the pathogenesis and multimodal management through the use of targeted therapy is particularly well written.
The strength of the manuscript is on the pathogenesis as well as the multimodal therapy. Although a minor criticism is that surgical therapy is not extensively covered in management of gallbladder cancer. Surgical treatment remains the only curative therapy for early stage gallbladder cancer. The authors may benefit from expanding on the use of surgical techniques including laparoscopic and robotic surgeries which are arising in frequency.
Dear Reviewer 3,
Thank you very much for your helpful advice! We have intensively discussed your recommendation in our author group, but since this special issue already contains a well-written and detailed review of the surgical therapy of GBC and since the other reviewers and the handling editor have suggested to focus on the pharmacotherapy, we have kept the surgical therapy section in its original form.
We hope you can agree with our arguments!
